# Regional practice variation in induction of labor in the Netherlands: Does it matter? A multilevel analysis of the association between induction rates and perinatal and maternal outcomes

Pien Offerhaus[1]*, Tamar M. van Haaren-Ten Haken[1], Judit K. J. Keulen[1], Judith D. de Jong[2,3], Anne E. M. Brabers[2], Corine J. M. Verhoeven[4,5,6,7,8,9], Hubertina C. J. Scheepers[10], Marianne Nieuwenhuijze[1,11]

1 Research Centre for Midwifery Science, Zuyd University, Maastricht, the Netherlands, 2 Nivel–Netherlands Institute for Health Services Research, Utrecht, The Netherlands, 3 Department of Health Services Research, Care and Public Health Research Institute, Maastricht University, Maastricht, The Netherlands, 4 Department of Midwifery Science, Amsterdam University Medical Centre (UMC), Vrije Universiteit Amsterdam, Amsterdam, the Netherlands, 5 Midwifery Academy Amsterdam Groningen, Inholland, Amsterdam, the Netherlands, 6 Amsterdam Public Health, Quality of Care, Amsterdam, the Netherlands, 7 Department of General Practice & Elderly Care Medicine, University Medical Center Groningen, University of Groningen, Groningen, the Netherlands, 8 Division of Midwifery, School of Health Sciences, University of Nottingham, Nottingham, United Kingdom, 9 Department of Obstetrics and Gynaecology, Maxima Medical Centre, Veldhoven, the Netherlands, 10 Department of Obstetrics and Gynecology, GROW School for Oncology and Reproduction, Maastricht University Medical Center, Maastricht, the Netherlands, 11 Maastricht University, Care and Public Health Research Institute, Maastricht, the Netherlands

* pien.offerhaus@zuyd.nl

## Abstract

### Background

Practice variation in healthcare is a complex issue. We focused on practice variation in induction of labor between maternity care networks in the Netherlands. These collaborations of hospitals and midwifery practices are jointly responsible for providing high-quality maternity care. We explored the association between induction rates and maternal and perinatal outcomes.

### Methods

In a retrospective population-based cohort study, we included records of 184,422 women who had a singleton, vertex birth of their first child after a gestation of at least 37 weeks in the years 2016–2018. We calculated induction rates for each maternity care network. We divided networks in induction rate categories: lowest (Q1), moderate (Q2-3) and highest quartile (Q4). We explored the association of these categories with unplanned caesarean sections, unfavorable maternal outcomes and adverse perinatal outcomes using descriptive statistics and multilevel logistic regression analysis corrected for population characteristics.

**Data Availability Statement:** "Data cannot be shared publicly because the perinatal register Perined does not allow public sharing of data that contain potentially identifying or sensitive information without women's consent. Data are available from Perined (contact via info@perined. nl) for researchers who meet the criteria for access to confidential data. The dataset underlying the results presented in this study is available from Perined (WWW.Perined.nl) on request. The current study is registered at Perined as number 20.09. Data can be requested by sending an email to info@Perined.nl or by filling in the application form 'Aanvraagformulier gegevens' that can be found on the webpage https://www.perined.nl/onderwerpen/ onderzoek/gegevensaanvragen."

**Funding:** This study was supported by a grant from the Netherlands Organisation for Health Research and Development (www.ZonMw.nl) grant no. 543003312. The funder had no role in study design, data collection and analysis, decision to publish, or preparation of the manuscript.

**Competing interests:** The authors have declared that no competing interests exist.

**Abbreviations:** CI, Confidence interval; IOL, Induction of labor; MCN, Maternity care network (*Verloskundig samenwerkings verband*; NICU, Neonatal intensive care unit; NTSV, Nulliparous term singleton vertex (pregnancy); SD, Standard deviation; SES, Social-economic status; SGA, Small for gestational age.

## Findings

The induction rate ranged from 14.3% to 41.1% (mean 24.4%, SD 5.3). Women in Q1 had fewer unplanned caesarean sections (Q1: 10.2%, Q2-3: 12.1%; Q4: 12.8%), less unfavorable maternal outcomes (Q1: 33.8%; Q2-3: 35.7%; Q4: 36.3%) and less adverse perinatal outcomes (Q1: 1.0%; Q2-3: 1.1%; Q4: 1.3%). The multilevel analysis showed a lower unplanned caesarean section rate in Q1 in comparison with reference category Q2-3 (OR 0.83; p = .009). The unplanned caesarean section rate in Q4 was similar to the reference category. No significant associations with unfavorable maternal or adverse perinatal outcomes were observed.

## Conclusion

Practice variation in labor induction is high in Dutch maternity care networks, with limited association with maternal outcomes and no association with perinatal outcomes. Networks with low induction rates had lower unplanned caesarean section rates compared to networks with moderate rates. Further in-depth research is necessary to understand the mechanisms that contribute to practice variation and the observed association with unplanned caesarean sections.

## Introduction

Practice variation in healthcare is a complex issue. Although some variation can be expected in relation to population differences or patient preferences, it is clear that other factors such as institutional mechanisms within organizations [1], individual practice styles of professionals and scientific uncertainty [2, 3] also contribute to practice variations. Variations are unwarranted if they cannot be explained by type or severity of illness, patient risk factors or patient preferences [2, 4, 5]. Practice variation may indicate insufficient quality of care as a result of underuse or overuse of interventions.

In maternity care, large practice variations in childbirth interventions have been reported, both between countries [6–8] and within countries [9–12]. This indicates that maternity care interventions may be performed 'too little, too late' or 'too much, too soon' [13]. Factors such as insufficient accessibility of care on the one hand or over-medicalization of normal physiological childbirth on the other hand, can result in underuse or overuse of interventions in maternity care. Both underuse and overuse may cause harm for the mother or the baby [13]. Reducing unwarranted practice variation can be a strategy for enhancing quality of maternity care and realizing optimal maternal and perinatal outcomes.

Practice variation in maternity care is an issue that also needs attention and further exploration in the Netherlands [11]. In the research project VAriation in Labor InDuction (VALID) we focus on practice variation in one intervention, namely induction of labor (IOL) in the Netherlands. In the VALID project we explore the different mechanisms that influence decision-making on IOL, how these mechanisms contribute to practice variation, what variation in IOL is unwarranted and how unwarranted variation may be reduced. Our focus on IOL has several reasons. First, in the last decades the use of IOL has increased in the Netherlands—as in many high income countries—to 24.4% of all term singleton births, affecting almost 40,000 women in 2019 [14]. Second, in a relatively homogenous group of women, IOL rates varied from less than 10% to over 40% in Dutch hospitals in 2012 [15]. Third, IOL is a major intervention in the course of a pregnancy, which should be performed only when the expected

benefits outweigh its potential harms. Although randomized controlled trials show some advantages for maternal or perinatal health in favor of IOL compared to expectant management in specific situations such as mild pre-eclampsia or a gestational age above 41 weeks [16, 17], in other situations the evidence supporting IOL is weak or conflicting [18–21]. In observational studies, IOL is associated with less favorable outcomes such as an increased need for epidural analgesia and more unplanned caesarean sections [22–26]. In short, risks and benefits of IOL in the term period are not always clear. Furthermore, a metasynthesis of qualitative research on women's experiences indicated that decision-making on IOL as well as undergoing an IOL can be challenging experiences for women [27]. In the Netherlands, an IOL is often preceded by a referral from midwife-led primary care to obstetrician-led secondary care. Such a referral in itself may affect women's experiences [28, 29].

It is unclear what impact practice variation in IOL rates has on maternal and perinatal outcomes. The objective of this substudy of the VALID project therefore is to explore the potential impact of this variation on maternal and perinatal outcomes. The results of the study can contribute to the reflection on the use of IOL in the Netherlands and internationally.

## Methods

We performed a population-based retrospective cohort study in the Netherlands using the Dutch national perinatal database Perined.

### Setting

In the Netherlands, almost 90% of all women start antenatal care in midwife-led primary care in the community [14]. As long as pregnancy and birth develop normally, women continue to receive midwife-led primary care and can opt for a birth at home, in a hospital or birth center, attended by their own midwife. In case of pregnancy or birth complications or increased risks, women are referred to obstetrician-led secondary care and the birth will take place in the hospital. Obstetrical interventions such as IOL are only available in obstetrician-led secondary care in the hospital, and require a referral to obstetrician-led care if a woman is still in midwife-led primary care [30]. This implies that in most cases both primary care midwives and secondary care staff members (obstetricians, residents and hospital-based midwives) are involved in the chain of decisions that leads to an advice for an IOL to individual women.

Midwifery practices and hospitals are organized regionally in maternity care networks (MCN). Next to midwives and obstetricians, these networks often other disciplines such as nurses, maternity care assistants and pediatricians [31]. In 2018, there were 77 MCNs in the Netherlands. Most MCNs are organized around one hospital and the midwifery practices in the same region. Usually, this hospital provides secondary obstetric and neonatal care with a Neonatal Intensive Care Unit (NICU) level 1 and 2. If the hospital is a tertiary (university) center, a NICU up to level 4 is available. Only ten MCNs have a tertiary center in their network, the other MCNs refer to these tertiary centers if necessary. The size of MCNs in terms of the number of midwives and obstetricians involved, varies from around 30 to 120 professionals and depends on the number of births in that region. An MCN is jointly responsible for providing high-quality maternity care in the region [32]. Within these MCNs, collaboration and reasons for referral are established between primary and secondary care, usually based on national guidelines and/or the national referral list for obstetric care [33]. Qualitative research shows that mechanisms within MCNs such as the interdisciplinary collaboration, local protocols, beliefs and attitudes towards childbirth, and women's involvement, influence decisions in maternity care [34, 35]. Therefore, MCNs are the appropriate unit of analysis for investigating regional practice variation in IOL in Dutch maternity care.

## Database and variables

The Perined database includes data from medical records of almost all births in the Netherlands, routinely collected in separate registers per maternity care professionals (midwives, general practitioners, obstetricians and pediatricians) and combined into one national database [14, 36]. For our study, we included a group of women with a relatively low risk for severe pregnancy complications. We selected all records in the years 2016–2018 of women who had a singleton, vertex birth of their first child (nulliparous) at a gestational age of at least 37 weeks (term): the so-called NTSV group. This NTSV group may be considered a relatively homogeneous study population and is therefore suitable to investigate regional practice variation [15]. Records with missing data on gestational age, parity, or vertex/non-vertex presentation at birth were not included.

For every record in the Perined database, the identity of the MCN is anonymously available for analysis. For each MCN, the presence of a NICU (level 4) in its own hospital or not is also registered (NICU availability). Perined assigns a record to a MCN based on the hospital of birth or, in case of a homebirth, based on the collaboration with local hospitals of the midwifery practice that attended the birth. Records in the Perined database with a missing MCN code were not included.

We collected information on the start of labor (spontaneous, induction of labor, caesarean section planned before the onset of labor) and the various obstetrical techniques used to induce labor (amniotomy, Foley or balloon catheter, prostaglandins or oxytocin). Membrane sweeping was not considered a method of induction of labor in this study.

Furthermore, we collected maternal, pregnancy, birth and perinatal characteristics as available in the Perined database. We collected information on maternal age, social economic status (SES) based on the four digits of the postal code and ethnic background (Dutch or non-Dutch). We also collected information on pregnancy complications (diabetes; suspected abnormal fetal growth: small for gestational age or large for gestational age; reduced fetal movements; hypertensive disorders) and on referrals from midwife-led primary care to obstetrician-led secondary care during antenatal care and during labor. Birth characteristics were: planned and actual place of birth, augmentation of labor with oxytocin, methods of pain relief (opioids or epidural anesthesia), birth mode (spontaneous vaginal, assisted vaginal, planned caesarean section before the onset of labor, unplanned caesarean section), postpartum blood loss > 1000 ml, and perineal trauma (3rd or 4th degree perineal tear; episiotomy). Perinatal characteristics were birthweight, Apgar score at 5 minutes, admission to a NICU in a tertiary center, perinatal mortality up to the 7th day after birth, and serious or lethal congenital malformations. A birthweight under the 3rd percentile was considered as small for gestational age, and above the 97th percentile as large for gestational age. Macrosomia was defined as a birthweight above 4500g.

## Analysis

**Practice variation.** IOL rates in the NTSV group were calculated per MCN, with the total number of women in the NTSV group per MCN as denominator. Mean IOL rate and the range in IOL rates in MCNs were calculated. We performed case-mix correction for available socio-demographic factors that are associated with maternal health and therefore may have impact on the IOL rate in MCNs based on population characteristics: women's social economic status (SES) and ethnic background. Lifestyle factors such as BMI and smoking were not available. Factors that may have variable impact per MCN on IOL decision are not included in the case-mix correction, since controlling for these factors might mask the practice variation that we aim to explore in this project. Therefore, we did not include maternal age,

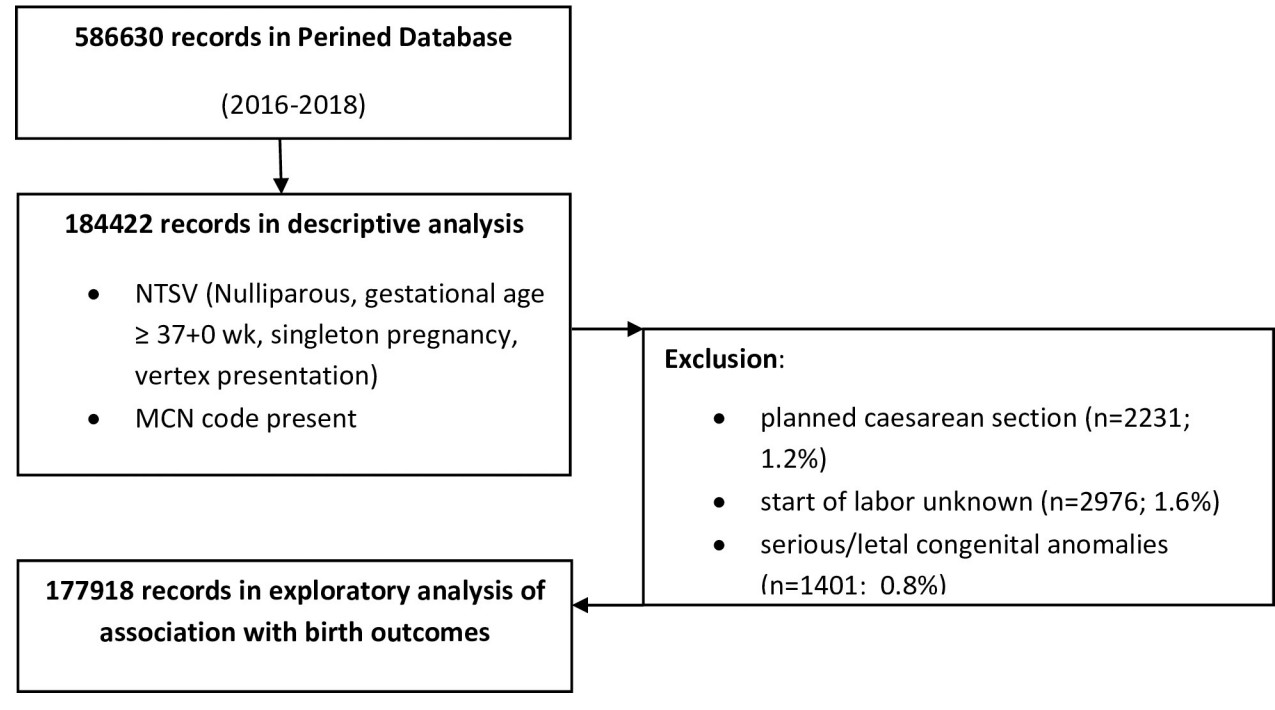

**Fig 1. Flowchart of the study population.**

birthweight, planned place of birth and NICU availability. We categorized MCNs, based on their ranking after case-mix correction. MCNs with a ranking in the lowest quartile (Q1) were categorized as having a low IOL rate and in the highest quartile (Q4) as having a high IOL rate. The other MCNs (Q2 and Q3) were categorized as having a moderate IOL rate.

**Association with outcomes.**   Our aim was to explore the association of a low or high IOL rate in MCNs with maternal and perinatal outcomes. Outcomes of interest were mode of birth (spontaneous vaginal birth, assisted vaginal birth or unplanned caesarean section), blood loss >1000 ml, 3rd/4th perineum tear or episiotomy, and adverse perinatal outcomes (perinatal mortality, low 5 minute Apgar score <4 and <7, and NICU admission within 24 hours). We defined a dichotomous maternal composite outcome describing unfavorable maternal outcomes, combining assisted vaginal birth or caesarean section, blood loss > 1000 ml, 3rd or 4th degree sphincter lesion. Adverse perinatal outcomes were combined in a composite, including perinatal mortality < 7 days, 5 minute Apgar score <4 and NICU admission.

We excluded records of the small group of women with a caesarean section planned before onset of labor (1.2%). Records in which Perined registered the birth of a child born with severe or lethal congenital anomalies were excluded as well (see Fig 1 for a flowchart).

We explored the associations between IOL categories and the outcomes of interest with descriptive statistics (Chi Square). We further explored these associations with multilevel logistic regression analysis to control for possible confounding by population characteristics that may explain variation in outcomes. We analyzed unplanned caesarean sections, the maternal composite outcome and the perinatal composite outcome in separate models. The independent variable in each model was the MCN category for IOL rates (Q1: low, Q2/Q3: moderate, Q4: high). The moderate IOL rate group was the reference category. We used multilevel fixed effects models to take into account clustering of women within MCNs. In the maternal models, we controlled forward stepwise for sociodemographic confounders (SES and ethnic background, maternal age) and pregnancy characteristics (presence of pregnancy complications,

macrosomia >4500g). Presence of pregnancy complications (diabetes; suspected abnormal fetal growth: small for gestational age or large for gestational age; reduced fetal movements; hypertensive disorders) were combined in a dichotomous composite confounding variable. In the perinatal model we controlled forward stepwise for sociodemographic confounders (SES and ethnic background; maternal age), NICU availability in the MCN, and pregnancy characteristics (presence of pregnancy complications, SGA). In none of the models we controlled for gestational age, since we consider this as an outcome of IOL rather than a characteristic in this study and therefore not as a confounding factor. We also did not control for level of care (midwife-led primary or obstetrician-led secondary care) at start of labor or place of birth, as we considered this partly an outcome of the maternity care process within the MCNs that might be related to their IOL rate [37]. Descriptive analysis was performed in SPSS version 25 and multilevel regression analysis using R version 4.0.1. (www.R-project.org).

### Ethical approval

The Medical Ethical Committee of Zuyderland-Zuyd University confirmed that ethical approval was not required for his study according to the Dutch legislation and regulations (reference METCCZ20210008). Consent was given by Perined for anonymous data analysis at MCN level.

## Results

### Description of the current variation

In the study period, the Perined database contained 586,630 records of 77 MCNs; 184,422 of these records concerned women with a nulliparous, term, singleton, vertex pregnancy: the NTSV group. The mean number of NTSV women per MCN in this three-year period was 2416.4 (SD 1334.4; range 660–6183). Overall, 23.6% of these women experienced an IOL: 23.3% in 2016, and 23.8% in both 2017 and 2018. The observed IOL rate per MCN in the NTSV group ranged from 14.3% to 41.1%.

MCNs were assigned to the lowest quartile (Q1), moderate quartiles (Q2-3) and highest quartile (Q4). The IOL rate ranged from 18.0% in Q1 to 30.8% in Q4, and the rate for spontaneous start of labor ranged from 66.6% (Q4) to 77.8% (Q1) in the three categories (see Table 1). Overall, the planned caesarean section rate was low and ranged from 1.0% in Q1 to 1.4% in Q4. Various methods for performing an IOL were used. In all groups, around half of the IOLs started with a Foley catheter. In Q4, more IOLs were performed from the beginning of the term period until the gestational age of $41^{+6}$ weeks, compared to Q2-3 and Q1. In week 42, more IOLs were performed in Q1. In all three categories, IOL rates were highest in week 38 and 41.

Population characteristics in the three IOL rate categories showed no great differences (see S3 Table), although more women lived in a higher SES area in Q1, and more in a lower SES area in Q4. Pregnancy complications as registered in Perined did not show large differences over the three categories. As expected in this relatively healthy group of women, in most records no pregnancy complications were registered (Q1: 76.9%; Q2-3: 75.7%; Q4: 73.5%). In the lowest IOL rate category, a larger proportion of women received midwife-led primary care at the start of antenatal care (Q1: 94.0%; Q2-3: 93.3%; Q4: 90.5%) and a larger proportion of women stayed in midwife-led primary care until start of labor (Q1: 66.6%; Q2-3: 60.5%; Q4: 52.3%).

Neonatal characteristics showed small differences. In line with the lower IOL rate, gestational age was the highest in Q1 (Q1: 279.9 days (SD 8.5); Q2-3: 279.4 (SD 8.6); Q4: 278.6 (SD

**Table 1. NTSV start of labor characteristics in MCNs with low, moderate or high induction of labor rates.**

| | Q1 (low) n = 52979 | | | Q2-3 (moderate) n = 91638 | | | Q4 (high) n = 39805 | | |
|---|---|---|---|---|---|---|---|---|---|
| **Start of labor** | | | | | | | | | |
| Spontaneous | 41209 | 77.8% | | 67908 | 74.1% | | 26513 | 66.6% | |
| Induction of labor (IOL) | 9537 | 18.0% | | 21793 | 23.8% | | 12255 | 30.8% | |
| Planned caesarean section | 535 | 1.0% | | 1139 | 1.2% | | 557 | 1.4% | |
| Unknown | 1698 | 3.2% | | 798 | 0.9% | | 480 | 1.2% | |
| **Method IOL** | | % within NTSV | % within IOL | | % within NTSV | % within IOL | | % within NTSV | % within IOL |
| Amniotomy only | 1301 | 2.5% | 13.6% | 3381 | 3.7% | 15.5% | 1013 | 2.5% | 8.3% |
| Prostaglandins | 1468 | 2.8% | 15.4% | 2672 | 2.9% | 12.3% | 1814 | 4.6% | 14.8% |
| Oxytocin | 2281 | 4.3% | 23.9% | 5856 | 6.4% | 26.9% | 3144 | 7.9% | 25.7% |
| Foley | 4487 | 8.5% | 47.0% | 9884 | 10.8% | 45.4% | 6284 | 15.8% | 51.3% |
| Total | 9537 | 18.0% | 100% | 21793 | 23.8% | 100% | 12255 | 30.8% | 100% |
| **IOL in week** | | % within NTSV | % within IOL | | % within NTSV | % within IOL | | % within NTSV | % within IOL |
| $37^{+0}$–$37^{+6}$ | 1206 | 2.3% | 12.6% | 2724 | 3.0% | 12.5% | 1505 | 3.8% | 12.3% |
| $38^{+0}$–$38^{+6}$ | 2034 | 3.8% | 21.3% | 4694 | 5.1% | 21.5% | 3172 | 8.0% | 25.9% |
| $39^{+0}$–$39^{+6}$ | 1633 | 3.1% | 17.1% | 3851 | 4.2% | 17.7% | 2171 | 5.5% | 17.7% |
| $40^{+0}$–$40^{+6}$ | 1386 | 2.6% | 14.5% | 3376 | 3.7% | 15.5% | 1737 | 4.4% | 14.2% |
| $41^{+0}$–$41^{+6}$ | 2207 | 4.2% | 23.1% | 5671 | 6.2% | 26.0% | 3271 | 8.2% | 26.7% |
| $\geq 42^{+0}$ | 1071 | 2.0% | 11.2% | 1477 | 1.6% | 6.7% | 399 | 1.0% | 3.1% |
| Total | 9537 | 18.0% | 100% | 21793 | 23.8% | 100% | 12255 | 30.8% | 100.0% |

Missings in variables are not shown. Due to missings, numbers not always add up to the total number in Q1, Q2-3 and Q4. Percentages are calculated on valid numbers.

8.6); p <0.001). In Q1 the mean birthweight was also slightly higher (Q1: 3443g (SD 457); Q2-3: 3434g (SD 460); Q4: 3387g (SD 456); p < 0.001).

## Birth characteristics

Table 2 describes birth characteristics in the three IOL-rate categories. The pregnancy and birth care path varied between the three categories. At the time of birth, a higher proportion of women in Q1 received midwife-led primary care (Q1: 25.6%, Q2-3: 21.8%; Q4: 17.5%), and less received secondary or tertiary care (Q1: 74.4%, Q2-3: 78.2%; Q4: 82.5%). At the same time, more women in Q1 had experienced an intrapartum referral from primary to obstetrician-led secondary care (Q1: 44.5%; Q2-3: 40.4%; Q4: 36.7%). More women in Q1 gave birth at home (Q1: 10.0%; Q2-3: 9.6%; Q4: 6.2%). In line with this variation in care paths, the use of interventions during labor varied. Overall use of oxytocin use during labor was lower in Q1 (Q1: 50.9%; Q2-3: 53.6%; Q4: 60.0%), as was the use of pharmaceutical pain relief (no use of pain relief in Q1: 52.0%; Q2-3: 47.5%; Q4:40.2%).

## Association of variation in IOL rates with maternal and perinatal outcomes

In Table 3, the outcomes of interest are described. We observed that more women in Q1 had a spontaneous vaginal birth (Q1: 73.2%; Q2-3: 71.8%; Q4: 69.9%; p < 0.001) and less women gave birth by unplanned caesarean section (Q1: 10.2%; Q2-3: 12.1%; Q4: 12.8%; p < 0.001). Postpartum blood loss >1000 ml (Q1: 7.5%; Q2-3: 7.8%; Q4: 6.9%, p< 0.001) as well as a sphincter lesion (Q1: 3.9%; Q2-3: 3.5%; Q4: 2.9%; p <0.001) were more prevalent in Q1 in comparison with Q4. Less women in Q1 received an episiotomy (Q1: 31.1%; Q2-3: 34.4%; Q4: 38.8%; p < 0.001). The maternal composite (non-spontaneous birth, postpartum blood loss >1000 ml, 3rd/4th degree tear) showed a lower prevalence in Q1 than in Q4 (Q1: 33.8%; Q2-3: 35.7%; Q4: 36.3%, p < 0.001).

**Table 2. Birth characteristics in MCNs with low, moderate or high induction of labor rates.** (NTSV group, excluding planned CS/congenital anomalies).

| | Q1 (low) n = 50419 | | Q2-3 (moderate) n = 89139 | | Q4 (high) n = 38360 | |
|---|---|---|---|---|---|---|
| **Birth characteristics** | | | | | | |
| **Place and level of care at birth** | | | | | | |
| Birth in primary midwife-led care | 12857 | 25.6% | 19350 | 21.8% | 6670 | 17.5% |
| *primary care, home* | 5026 | 10.0% | 8538 | 9.6% | 2372 | 6.2% |
| *primary care, birth center* | 2101 | 4.2% | 1056 | 1.2% | 425 | 1.1% |
| *primary care, place unknown* | 999 | 2.0% | 570 | .6% | 272 | .7% |
| *primary care, hospital* | 4731 | 9.4% | 9186 | 10.3% | 3601 | 9.4% |
| Birth in secondary/tertiary care, hospital | 37343 | 74.4% | 69442 | 78.2% | 31459 | 82.5% |
| *after referral from primary care during labor* | 22330 | 44.5% | 35899 | 40.4% | 14002 | 36.7% |
| **Augmentation/oxytocin first stage** | 24262 | 50.9% | 46854 | 53.6% | 22461 | 60.0% |
| *after spontaneous start* | 17802 | 45.9% | 31472 | 47.7% | 13044 | 51.2% |
| *after IOL by prostaglandins* | 959 | 68.1% | 1695 | 64.4% | 1161 | 67.1% |
| *after IOL by Foley* | 3436 | 80.8% | 7906 | 81.3% | 5155 | 84.0% |
| *after IOL by oxytocin* | 2065 | 100.0% | 5781 | 100.0% | 3101 | 100.0% |
| **Pharmaceutical pain relief first stage** | | | | | | |
| none | 26199 | 52.0% | 42362 | 47.5% | 15430 | 40.2% |
| opioid | 5395 | 10.7% | 10374 | 11.6% | 5235 | 13.6% |
| epidural | 15410 | 30.6% | 32163 | 36.1% | 15738 | 41.0% |
| other/unknown | 3415 | 6.8% | 4240 | 4.8% | 1957 | 5.10% |

Missings in variables are not shown. Due to missings, numbers not always add up to the total number in Q1, Q2-3 and Q4. Percentages are calculated on valid numbers.

The perinatal composite (perinatal mortality, AS <4, NICU) showed a lower prevalence in Q1 (Q1: 1.0%; Q2-3: 1.1%; Q4: 1.3%, p = 0.005) in the descriptive analysis. Admittance to a NICU within 24 hours (Q1: 0.8%: Q2-3: 0.8%; Q4: 1.1%, p < 0.001) contributed most to this difference. The other elements in the perinatal composite showed no significant differences between the IOL categories in this analysis.

Table 4 presents the outcomes of the exploratory multilevel analysis for maternal and perinatal outcomes. Q2-3 was the reference category in all models. For the outcome unplanned caesarean section, the crude analysis showed a lower risk in the low IOL rate category Q1 (odds ratio (OR) 0.84; 95% CI 0.81–0.87; p <0.001), and a higher risk in the high IOL rate category Q4 (OR 1.07; 95% CI 1.03–1.11; p <0.001) in comparison with Q2-3. In the multilevel model, the lower risk remained statistically significant for Q1, but for Q4 the difference with the reference category was no longer statistically significant. Controlling for sociodemographic confounders and for pregnancy characteristics and macrosomia did not alter this result. In the final model, the OR for Q1 was 0.83 (95% CI 0.72–0.95; p = 0.009) after controlling for all confounders, suggesting a lower risk for unplanned caesarean section in MCNs with a low IOL rate in comparison with the reference category. For the high IOL rate category Q4, the multilevel analysis showed no difference in risk for an unplanned caesarean section, in comparison with the reference category.

The crude OR for the unfavorable maternal outcomes composite was significantly lower for Q1 (Q1: OR 0.92; 95% CI 0.90–0.94; p < 0.001). For Q4 no significant difference with the reference category was observed (Q4: OR 1.02; 95% CI 0.998–1.05; p = 0.073). In the multilevel analysis, we found no statistically significant association with the IOL rate categories in any model.

For the perinatal composite, neither the crude analysis nor the multilevel models showed any statistically significant association with the IOL rate categories.

**Table 3. Maternal and perinatal outcomes in MCNs with low, moderate or high induction of labor rates.** (NTSV group, excluding planned caesarean section/congenital anomalies).

| | Q1 (low) n = 50419 | | Q2-3 (moderate) n = 89139 | | Q4 (high) n = 38360 | |
|---|---|---|---|---|---|---|
| | *n* | *% (95% CI)* | *n* | *% (95% CI)* | *n* | *% (95% CI)* |
| **Mode of birth** | | | | | | |
| Spontaneous vaginal** | 36883 | 73.2 (72.8–73.5) | 63972 | 71.8 (71.5–72.1) | 26825 | 69.9 (69.5–70.4) |
| Assisted vaginal** | 7329 | 14.5 (14.2–14.8) | 13513 | 15.2 (14.9–15.4) | 6200 | 16.2 (15.8–16.5) |
| Unplanned caesarean section** | 5148 | 10.2 (9.9–10.5) | 10742 | 12.1 (11.8–12.3) | 4891 | 12.8 (12.4–13.1) |
| unknown | 1059 | 2.1 (2.0–2.2) | 912 | 1.0 (1.0–1.1) | 444 | 1.2 (1.1–1.3) |
| **Maternal outcomes** | | | | | | |
| Postpartum blood loss (PPH) > 1000 cc** | 3776 | 7.5 (7.3–7.7) | 6989 | 7.8 (7.7–8.0) | 2653 | 6.9 (6.9–7.2) |
| 3rd/4th degree perineal tear** | 1957 | 3.9 (3.7–4.1) | 3085 | 3.5 (3.3–3.6) | 1127 | 2.9 (2.8–3.1) |
| Episiotomy** | 15688 | 31.1 (30.7–31.5) | 30707 | 34.4 (34.1–34.8) | 14880 | 38.8 (38.3–39.3) |
| Unfavorable maternal outcomes composite** (non-spontaneous birth, PPH, 3rd/4th degree tear) | 16718 | 33.8 (33.4–34.2) | 31571 | 35.7 (35.4–36.1) | 13763 | 36.3 (35.8–36.8) |
| **Perinatal outcomes** | | | | | | |
| Apgar score 5 min <7# | 946 | 1.9 (1.8–2.0) | 1517 | 1.7 (1.6–1.8) | 687 | 1.8 (1.7–1.9) |
| Apgar score 5 min < 4# | 170 | 0.3 (0.3–0.4) | 294 | 0.3 (0.3–0.4) | 116 | 0.3 (0.3–0.4) |
| Perinatal mortality up to day 7# | 66 | 0.13 (0.10–0.17) | 130 | 0.15 (0.12–0.17) | 42 | 0.11 (0.08–0.15) |
| NICU admission** | 402 | 0.8 (0.7–0.8) | 756 | 0.8 (0.8–0.9) | 404 | 1.1 (1.0–1.2) |
| Adverse perinatal composite* (mortality, AS<4, NICU) | 518 | 1.0 (0.9–1.1) | 980 | 1.1 (1.1–1.2) | 482 | 1.3 (1.1–1.4) |

Missings in variables are not shown. Due to missings, numbers not always add up to the total number in Q1, Q2-3 and Q4. Percentages are calculated on valid numbers.

Outcomes were tested statistically (Chi Square)

CI: confidence interval

# no statistically significant difference, $p > 0.05$

* statistically significant difference, $p < 0.05$

** statistically significant difference, $p < 0.001$

## Discussion

We observed high practice variation between maternity care networks (MCNs) in IOL rates in a relatively homogenous group of nulliparous women with a term singleton vertex birth (NTSV group). We also observed that the association of these IOL rates with maternal or perinatal outcomes is limited. In a multilevel logistic regression analysis, the unplanned caesarean section rate was lower in the MCNs with a low IOL rate compared to MCNs in the moderate group, but in MCNs with a high IOL rate this outcome was comparable with the moderate group. We found no other significant associations between IOL rates and perinatal or maternal outcomes in this multilevel analysis.

Our exploratory analysis suggests a lower number of unplanned caesareans in the NTSV group in MCNs with a low IOL rate, and no impact on the other investigated outcomes. This lower caesarean section rate in our observational study seems in contrast with several RCTs that report no significant differences in caesarean sections when IOL is compared with expectant management in case of specific pregnancy complications [16–19]. The results of the ARRIVE trial [20] even suggests that IOL at 39 weeks gestation reduced caesarean sections in comparison with expectant management in a study population of nulliparous women with

**Table 4. Associations of low and high rates in induction of labor in MCNs with primary outcomes, in comparison with MCNs with a moderate induction of labor rate in (multilevel modelling).**

| | OR (95% CI) Q1 (low) | p-value | OR (95% CI) Q4 (high) | p-value | variance |
|---|---|---|---|---|---|
| **UNPLANNED CAESAREAN SECTION** | | | | | |
| *crude OR* | | | | | |
| (no modelling) | 0.84 (0.81–0.87) | < 0.001 | 1.07 (1.03–1.11) | < 0.001 | NA |
| *1.Empty model* | | | | | |
| (fixed effect multilevel, no confounders) | 0.83 (0.72–0.95) | 0.006 | 1.02 (0.89–1.17) | 0.777 | 0.053 |
| *2.Sociodemographic confounders* | | | | | |
| **2a**: model 1 + SES and Ethnic background | 0.83 (0.72–0.95) | 0.008 | 1.01 (0.88–1.16) | 0.885 | 0.057 |
| **2b**: model 1 + SES and Ethnic background + maternal age | 0.82 (0.71–0.95) | 0.007 | 1.01 (0.88–1.17) | 0.874 | 0.061 |
| *3.Sociodemographic + pregnancy characteristics* | | | | | |
| **3a**: model 2b + pregnancy complications | 0.83 (0.72–0,96) | 0.010 | 0.99 (0.88–1.16) | 0.896 | 0.062 |
| **3b**: model 2b + macrosomia (birthweight ≥ 4500g) | 0.82 (0.71–0.95) | 0.006 | 1.02 (0.89–1.18) | 0.747 | 0.062 |
| **3c**: model 2b + pregnancy complications + macrosomia | 0.83 (0.72–0.95) | 0.009 | 1.00 (0.87–1.16) | 0.967 | 0.062 |
| **MATERNAL COMPOSITE** | | | | | |
| *crude OR* | | | | | |
| (no modelling) | 0.92 (0.90–0.94) | < 0.001 | 1.02 (0.998–1.05) | 0.073 | NA |
| *1.Empty model* | | | | | |
| (fixed effect multilevel, no confounders) | 0.93 (0.85–1.01) | 0.103 | 1.03 (0.94–1.12) | 0.526 | 0.023 |
| *2.Sociodemographic confounders* | | | | | |
| **2a**: model 1 + SES and Ethnic background | 0.93 (0.85–1.01) | 0.087 | 1.03 (0.94–1.12) | 0.513 | 0.023 |
| **2b**: model 1 + SES and Ethnic background + maternal age | 0.92 (0.84–1.01) | 0.068 | 1.03 (0.94–1.13) | 0,473 | 0.024 |
| *3.Sociodemographic + pregnancy characteristics* | | | | | |
| **3a**: model 2b + pregnancy complications | 0.92 (0.84–1.01) | 0.085 | 1.02 (0.94–1.12) | 0.608 | 0.025 |
| **3b**: model 2b + macrosomia (birthweight ≥ 4500g) | 0.92 (0.84–1.01) | 0.069 | 1.04 (0.95–1.14) | 0.387 | 0.024 |
| **3c**: model 2b + pregnancy complications + macrosomia | 0.92 (0.84–1.01) | 0.085 | 1.03 (0.94–1.13) | 0.508 | 0.025 |
| **NEONATAL COMPOSITE** | | | | | |
| *crude OR* | | | | | |
| (no modelling) | 0.93 (0.84–1.04) | 0.210 | 1.14 (1.03–1.28) | 0.016 | NA |
| *1.Empty model* | | | | | |
| (fixed effect multilevel, no confounders) | 0.999 (0.67–1.50) | 0.998 | 0.97 (0.64–1.47) | 0.869 | 0.501 |
| *2.Sociodemographic confounders* | | | | | |
| **2a**: model 1 + SES and Ethnic background | 1.01 (0.67–1.52) | 0.967 | 0.96 (0.63–1.47) | 0.864 | 0.500 |
| **2b**: model 1 + SES and Ethnic background + maternal age | 1.01 (0.67–1.52) | 0.971 | 0.99 (0.63–1.47) | 0.870 | 0.494 |
| *3.Sociodemographic including NICU availability* | | | | | |
| **3**: model 2b + NICU availability | 0.96 (0.76–1.22) | 0.748 | 0.86 (0.67–1.10) | 0.220 | 0.124 |
| *4.Sociodemographic + pregnancy characteristics* | | | | | |
| **4a**: model 3 + pregnancy complications | 0.96 (0.76–1.22) | 0.757 | 0.86 (0.67–1.10) | 0.218 | 0.124 |
| **4b**: model 3 + SGA (birthweight <p3) | 0.97 (0.77–1.22) | 0.761 | 0.86 (0.67–1.09) | 0.215 | 0.123 |
| **4c**: model 3 + SGA + pregnancy complications | 0.97 (0.77–1.22) | 0.769 | 0.86 (0.67–1.09) | 0.212 | 0.123 |

OR: odds ratio; CI: confidence interval

uncomplicated pregnancies. However, the ARRIVE trial reported a caesarean section rate of 18.6% in the IOL group vs. 22.2% in the expectant management group, which is considerably higher than the caesarean section rate observed in our study groups. Furthermore, the ARRIVE trial might suffer from selection bias [38]. It is therefore questionable whether the results of the ARRIVE trial can be generalized to the Dutch setting.

Altogether, the causality of the association we observed is unclear and requires further investigation. We cannot simply recommend MCNs to lower their use of IOL based on this observation. Firstly, we do not know what mechanisms explain the association we found. For instance, it may well be that an overall non-interventionist-approach in these MCNs explains both the low IOL rate and the lower number of unplanned CS. This is underpinned by our observation that the pregnancy and birth care process varied between the MCNs, in association with the IOL rates. On average, the provision of midwife-led primary care during birth was higher in the MCNs with low IOL rates. In addition, early term IOL before 39 weeks, and interventions during labor and birth were lower than in the MCNs with high IOL rates. Interestingly, in these MCNs also more women gave birth at home. Altogether, this suggests that a low IOL rate in a MCN represents a maternity care philosophy with a stronger tendency to a physiological, non—interventionist approach, probably both in professionals and in the women to whom they provide care. Secondly, a general recommendation to lower the IOL rate in a specific MCN might affect the decision-making in situations where IOL is the optimal treatment based on sound evidence. A MCN that considers changing their overall IOL rates needs to investigate on what indications the decision-making process on IOL might be optimized, taking available evidence and national guidelines into account.

Considering the limited association with outcomes, the high practice variation in IOL in the NTSV group warrants close attention. In the Netherlands, the general advice during the study period was—and still is–not to actively offer elective induction of labor to women [39]. Only if medical complications or risks are present, such as pregnancy hypertension or suspected fetal growth retardation induction of labor should be advized [39], or if pregnancy is prolonged [40]. It is unlikely that in our relatively homogenous study population, the observed variation in IOL can be fully explained by the prevalence of such pregnancy complications, especially since maternal and pregnancy characteristics in our descriptive analysis showed limited differences between the MCNs with a lower, moderate or higher IOL rate.

Instead, our study suggests that other mechanisms within the MCNs contribute to the variation. These mechanisms might be related to the organization or protocols within the MCNs, or to shared norms and beliefs [34, 41]. Various authors suggest that a perception of pregnancy and birth as predominantly physiological events or more as risky events, may influence maternity care professionals in their tendency to intervene [35, 42–45]. Especially in situations where the evidence base to perform an IOL is not very strong [21], such mechanisms may lead to variation between MCNs. For instance, studies on IOL such as the ARRIVE trial may influence beliefs and attitudes and may change the threshold to offer IOL, regardless of recommendations in national guidelines.

Although most studies on practice variation focus on professional decisions for medical treatment, the role of women should not be overlooked. Preferences of women are important, and variation should be found on the level of the patient rather than on the level of professionals or their organizations [46]. The literature on practice variation suggests that involvement of patient preferences by introducing shared decision-making may shift variation from the level of professionals and their organizations to variations on the individual level of patients. Some evidence supporting this suggestion is already available [47]. In recent years, the value of shared decision-making in maternity care in the Netherlands has been recognized and women are increasingly involved in decisions. Many women have strong preferences regarding IOL decisions, reflected in requests for elective IOL, or on preferences for expectant management in case of mild pregnancy complications [48]. However, research among women who recently gave birth in the Netherlands suggests that women's involvement in IOL decisions is not optimal. A substantial proportion of women who experienced an IOL, perceived at least some

pressure from professionals to accept this intervention [49], as was earlier described in other countries [50, 51].

Altogether, more in depth research is needed to understand the dynamics in professionals and their organizations and in women that contribute to the observed variation in IOL rates. Therefore, the next step in the VALID project will be to unravel the mechanisms that contribute to the observed practice variation, taking into account the social contexts of the MCNs and differences in population or policy. We use a sociological model for understanding practice variation that includes mechanisms in professionals and their local MCNs as well as in women, including the process of shared decision-making [52]. Insights in these mechanisms will help to define recommendations to optimize decision-making for IOL, and may reduce unwarranted practice variation in IOL.

## Limitations and strength

We performed our study with routinely collected data from the Perined database. This database gives limited information on several health determinants that may vary between MCNs, such as smoking and obesity. Therefore, we could not include these factors in our analysis and some confounding may still be present. However, we do not expect that correction for these factors would have much impact on the results of the multilevel models. Correction for other potential confounders (SES, maternal age, pregnancy complications) had only limited impact on the outcome of the modelling procedures.

Another limitation of the Perined database is that a straightforward registration of pregnancy complications, such as hypertension or reduced fetal movements, is not available. Perined computes the presence of these complications using various variables recorded in the separate registers. Some misclassification may occur, but we do not consider this as a source for important bias in our study, as this misclassification is not selective.

Despite these limitations, the Perined database offers the optimal opportunity to study practice variation within the Netherlands. Variation in IOL rates may be a result of complex patterns in MCNs, at the level of women, midwifery practices and hospitals. A major strength of the Perined database is that information on all these levels is available on a national level. Therefore, we were able to include the total NTSV population of the Netherlands in our study.

## Conclusion

Practice variation between MCNs in IOL rates in the NTSV group is high in the Netherlands, ranging from 14.3% to 41.1%. We observed no association between IOL rates and perinatal outcomes, and a limited association with maternal outcomes. MCNs with low IOL rates had lower unplanned caesarean section rates compared to MCNs with moderate IOL rates. MCNs with high IOL rates had similar unplanned caesarean section rates in comparison with MCNs with moderate IOL rates. More in depth research is necessary to understand the mechanisms that contribute to the observed high practice variation and to explain the observed association with unplanned caesarean sections. This may help to reduce unwarranted variation between MCN and improve quality of care.

## Supporting information

**S1 Table. Overview of nulliparous term singleton vertex (NTSV) population in MCNs 2016–2018.**
(DOCX)

**S2 Table. Start of labor per year in NTSV population.**
(DOCX)

**S3 Table. NTSV population and pregnancy characteristics in MCNs with low, moderate and high IOL rate.**
(DOCX)

**S1 Fig. MCN practice variation in IOL in nulliparous women with a term singleton pregnancy with a vertex presentation.**
(DOCX)

## Acknowledgments

We thank Lisa Broeders (Perined) for supporting data provision, and Marian Heins (NIVEL) for her support with the statistical analysis

## Author Contributions

**Conceptualization:** Pien Offerhaus, Tamar M. van Haaren-Ten Haken, Judit K. J. Keulen, Judith D. de Jong, Anne E. M. Brabers, Corine J. M. Verhoeven, Marianne Nieuwenhuijze.

**Formal analysis:** Pien Offerhaus.

**Writing – original draft:** Pien Offerhaus.

**Writing – review & editing:** Pien Offerhaus, Tamar M. van Haaren-Ten Haken, Judit K. J. Keulen, Judith D. de Jong, Anne E. M. Brabers, Corine J. M. Verhoeven, Hubertina C. J. Scheepers, Marianne Nieuwenhuijze.

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
