## [Decision Letter · Decision Letter 0]

21 Dec 2022

PONE-D-22-29662Regional practice variation in induction of labor in the Netherlands: does it matter? A multilevel analysis of the association between induction rates and perinatal and maternal outcomesPLOS ONE

Dear Dr. Offerhaus,

Thank you for submitting your manuscript to PLOS ONE. After careful consideration, we feel that it has merit but does not fully meet PLOS ONE’s publication criteria as it currently stands. Therefore, we invite you to submit a revised version of the manuscript that addresses the points raised during the review process.

We look forward to receiving your revised manuscript.

Kind regards,

David Desseauve, MD, MPH, PhD

Academic Editor

PLOS ONE

Journal Requirements:

Reviewers' comments:

Reviewer's Responses to Questions

**Comments to the Author**

1. Is the manuscript technically sound, and do the data support the conclusions?

Reviewer #1: Yes

Reviewer #2: Partly

2. Has the statistical analysis been performed appropriately and rigorously? 

Reviewer #1: Yes

Reviewer #2: No

3. Have the authors made all data underlying the findings in their manuscript fully available?

Reviewer #1: Yes

Reviewer #2: Yes

4. Is the manuscript presented in an intelligible fashion and written in standard English?

Reviewer #1: Yes

Reviewer #2: Yes

5. Review Comments to the Author

Reviewer #1: General comments

The objective of this Dutch population-based retrospective cohort study was to explore the association between IOL variations and maternal and perinatal outcomes. The authors focused on a homogeneous group: nulliparous women with a vertex singleton at term. They reported that IOL rates ranged from 14.3% to 41.1% in the Netherlands between 2016 and 2018 in 77 maternity care networks. They observed a lower risk of unplanned cesarean delivery in maternity care networks with low versus moderate IOL rates after controlling for maternal and neonatal cofunding factors and accounting for clustering in maternity care networks through multilevel logistic regression. They observed no difference between IOL rates and poor maternal or perinatal outcomes. Interestingly, interventions during labor (augmentation of labor with oxytocin, use of epidural analgesia, episiotomy) were lower in networks with low IOL rates than in networks with high IOL rates, suggesting a philosophy of maternity care based on a physiologic, noninterventionist approach.

The manuscript is original and well written. It covers a very interesting topic on obstetric interventions and their impact on adverse maternal and perinatal outcomes. Finally, the authors pointed out that women's preference may be the key to understanding variations in interventions. They pointed out that patient involvement through shared medical decisions is essential.

Nevertheless, there are some limitations. First, all results are presented without confidence intervals, which limits their interpretation. Second, the authors categorized maternity care networks according to their case-mix adjustment ranking in a logistic regression, but they did not explain why they used only two sociodemographic variables (social economic status and ethnicity) in their model. Thus, the categorization of low, moderate, or high IOL rate should be developed and clarified. In the multilevel logistic regression, there was no adjustment for gestational age. The authors considered gestational age not to be a cofactor but an outcome of IOL. However, poor neonatal outcomes are related to gestational age. In addition, the distribution of gestational age varied from the lowest to the highest IOL rates. There were significantly more IOLs performed for prolonged pregnancy (>42SA) in Q1 than in Q2-3 and Q4. Unfortunately, obesity was absent from the Perined database. Obesity may be a factor in IOL failure or adverse maternal outcomes. The authors discussed this limitation in the Discussion section.

Here are some minor comments

Abstract

Page 8 Lines 30-31 The objective is not clearly formulated

Page 9 Line 51-52 “Networks with low induction rates had lower unplanned caesarean section rates compared to networks with moderate or high rates”. Sentence should be rephrased as networks with low induction rates were not compared to networks with high rates.

Methods

Page 13 Line 149 : What did you mean by “birth characteristics were planned” ?

Page 14, Line 182- 183 “We analyzed unplanned caesarean sections, the maternal composite outcome and the perinatal composite outcome in separate models. » Why did you choose to analyze the association between IOL categories and unplanned cesarean deliveries and a maternal composite outcome which included cesarean delivery ?

Page 15 Line 188 Could you precise « pregnancy complications » ? Is it included small and large for gestational age ?

Page 15 Line 190 « NCIU availability » This variable was not described in the « database and variables » section. Could you provide clarification ?

Reviewer #2: The work presented here is very interesting and the variations in IOL is an important subject that needs to be better assessed.

My comments

Introduction : the introduction is quite long. Nevertheless, I don't feel that it reflects well the problematic that is the risk and benefit of IOL. Moreover it does not cite a very important paper on the subject that is the ARRIVE trial which significantly changed the current practice.

The introduction should better clarify the research question that is either to confirm that variation exist, which is my opinion is not such an important message and two what are the impact of such variations on maternal and neonatal outcomes which is indeed very interesting.

Methods & Results :

The reason for induction is not described. This is very important as it might significantly influence both maternal and neonatal outcomes. Similarly we lack information about maternal characteristics such as age or BMI which may influence outcomes. Although the authors controlled for such coufounding factors these should be clearly shown.

Table 3 Confidence interval are missing which impairs comparaison

Discussion The ARRIVE trial should be discussed

6. PLOS authors have the option to publish the peer review history of their article (what does this mean?). If published, this will include your full peer review and any attached files.

Reviewer #1: No

Reviewer #2: No

---

## [Author Response · Author response to Decision Letter 0]

13 Feb 2023

We would like to thank both reviewers for their time and effort to comment on our study. We address the remarks one by one for each reviewer. Line numbers in our response refer to the revised version with track changes. 

Response to reviewer 1

We thank the reviewer for the compliments on the article. 

• “all results are presented without confidence intervals” 

We added confidence intervals to Table 3 and 4, in which the outcomes of interest are presented and analysed. Table 1 and 2 present the description of the national study population. Since this information is purely descriptive, we do not think confidence intervals add much value and we did not calculate them.

• “[the authors] did not explain why they used only two sociodemographic variables (social economic status and ethnicity) in their model. Thus, the categorization of low, moderate, or high IOL rate should be developed and clarified.”

We performed case-mix correction for available socio-demographic factors that are associated with maternal health and therefore may have an impact on the IOL rate in MCNs based on population characteristics: women’s social economic status (SES) and ethnic background. Lifestyle factors that might have an impact the IOL rate such as BMI and smoking were not available in our data. Factors that may have variable impact per MCN on IOL decisions are not included in the case-mix correction, since controlling for these factors might mask the practice variation that we aim to explore in this project. Therefore, we did not include maternal age, birthweight, planned place of birth and NICU availability. We added this information on the case mix correction in line 168-174. 

• “The authors considered gestational age not to be a cofactor but an outcome of IOL. However, poor neonatal outcomes are related to gestational age. In addition, the distribution of gestational age varied from the lowest to the highest IOL rates. There were significantly more IOLs performed for prolonged pregnancy (>42GA) in Q1 than in Q2-3 and Q4.”

We did consider adjustment for gestational age. However, our population consists of women with term pregnancies. Within the term period, the association of gestational age with perinatal outcomes exists but is of limited magnitude, until a gestational age of 42 weeks. Since we considered gestational age also an outcome, we decided altogether not to include this in the multilevel models. 

We indeed observed a small difference in gestational age (Q1: 279.9 days; Q2-3: 279.4 days; Q4: 278.6 days) and somewhat more IOLs in Q1 (2.0% vs 1.6% in Q2-3 and 2.0% in Q4) for prolonged pregnancy. We hypothesize in the discussion that this is a result of different policies in Q1 and Q4. If more expectant management is practiced in a region, leading to less IOLs throughout the term period (as is shown in table 1), more women will reach a gestational age of 42 weeks and will receive an advice for labour induction. One could argue that having more prolonged pregnancies in a region might lead to less favourable neonatal outcomes, but we did not observe this in our models, even without controlling for gestational age. 

Minor remarks: 

• Lines 30-32 The objective is not clearly formulated

We changed the objective in: “We explored the association between induction rates and maternal and perinatal outcomes.”

• Line 51-53 “Networks with low induction rates had lower unplanned caesarean section rates compared to networks with moderate or high rates”. Sentence should be rephrased as networks with low induction rates were not compared to networks with high rates.

We removed “or high”.

• Line 155 : What did you mean by “birth characteristics were planned” ?

We added a colon to help the correct reading of this sentence (line 155): ‘ Birth characteristics were: planned and actual place of birth, …etc.’

• Line 195-196 “We analyzed unplanned caesarean sections, the maternal composite outcome and the perinatal composite outcome in separate models. » Why did you choose to analyze the association between IOL categories and unplanned cesarean deliveries and a maternal composite outcome which included cesarean delivery?

We decided to use this maternal composite, because we wanted to describe unfavorable maternal outcomes in one measure, to be able to perform our explorative analysis. In such a composite measure, unplanned caesarean section cannot be excluded. We also analyzed unplanned caesarean sections separately, because this is considered an important or primary maternal outcome in many studies. 

• Line 201 Could you precise « pregnancy complications » ? Is it included small and large for gestational age ?

We added the following description of ‘pregnancy complications’ to the manuscript (line 201-203) ‘Presence of pregnancy complications (diabetes; suspected abnormal fetal growth: small for gestational age or large for gestational age; reduced fetal movements; hypertensive disorders) were combined in a dichotomous composite confounding variable.’

• Line 205 « NCIU availability » This variable was not described in the « database and variables » section. Could you provide clarification?

In the method section, we added: ‘For each MCN, the presence of a NICU (level 4) in its own hospital or not is also registered (NICU availability)’ (line 141-142).

Response to reviewer 2

• “The work presented here is very interesting and the variations in IOL is an important subject that needs to be better assessed.”

We thank the reviewer for this compliment and for acknowledging the importance of this topic. 

• Introduction: I don't feel that [the introduction] reflects well the problematic that is the risk and benefit of IOL (##). Moreover, it does not cite a very important paper on the subject that is the ARRIVE trial which significantly changed the current practice (###).

The introduction should better clarify the research question (#) that is either to confirm that variation exist, which is my opinion is not such an important message and two what are the impact of such variations on maternal and neonatal outcomes which is indeed very interesting.

We react on the various underlined topics (#, ## and ###) in this remark 

# We better clarified in the introduction that our research is an explorative study, that aims to explore the association between IOL rates and maternal and perinatal outcomes (line 97): ‘The objective of this substudy of the VALID project therefore is to explore the potential impact of this variation on maternal and perinatal outcomes.’ 

## Our study is not on the risk or benefit of IOL, but on practice variation in IOL and it is part of a larger research project into practice variation, the VALID study. We presented some more information on the VALID study in lines 77-80. The introduction of the current substudy aims to explain why we focus on practice variation in IOL in the VALID project. In the final stage of the VALID study, we plan to discuss to what extent practice variation is warranted or unwarranted. Information on outcomes associated with this practice variation can contribute to this discussion. 

### In the introduction, we present some of the issues concerning IOL. One of the issues is that the evidence for the benefit or harm of IOL versus expectant management is not always clear. We refer in that context to the ARRIVE trial (reference 20, line 88), among other studies. We added a sentence (line 90-91): ‘In short, risks and benefits of IOL in the term period are not always clear.’

At this moment, the effects of the ARRIVE trial on the Dutch IOL policy are not clear. A national guideline on elective induction was published after publication of the ARRIVE trial, in which it is recommended not to actively offer elective induction to women with uncomplicated term pregnancies. We mention this guideline in the discussion section (reference 39, see lines 338-340). However, the publication of the ARRIVE trial generated some discussion in the Netherlands. It is possible that it influenced the individual attitudes and beliefs of care professionals or pregnant women – in both directions, either a more liberal or more restricted attitude towards IOL- , resulting in different regional policies towards IOL. We added a sentence in the discussion section to explain this (lines 351-352: “For instance, studies on IOL such as the ARRIVE trial may influence beliefs and attitudes and may change the threshold to offer IOL, regardless of recommendations in national guidelines.”)

This highlights the importance of the broader VALID project: to explore and understand the mechanisms of practice variation on IOL and the association with outcomes. 

• Methods & Results :

The reason for induction is not described. This is very important as it might significantly influence both maternal and neonatal outcomes. Similarly, we lack information about maternal characteristics such as age or BMI, which may influence outcomes. Although the authors controlled for such confounding factors these should be clearly shown.

The main reason for not describing the reasons for induction is that this information is not reliably available in the Perined dataset. Furthermore, ‘reason for induction’ is not a stable indicator for the health or complications in pregnancy. Reasons for induction are subject to regional variation within the Netherlands. This is our experience in the broader VALID project (publication in preparation). Recently other authors showed this also for the topic ‘fetal growth restriction’ (Marijnen et al. Practice variation in diagnosis, monitoring and management of fetal growth restriction in the Netherlands. Eur J Obstet Gynecol Reprod Biol. 2022, doi 10.1016/j.ejogrb.2022.07.021). 

We agree with the reviewer that pregnancy problems that might be reason for IOL may influence outcomes. We therefore used registered pregnancy complications as an indicator for maternal health in pregnancy. In this revised version, we explain our definition of pregnancy complications in lines 201-203. We added the presence of pregnancy complications as a confounder in the multilevel analysis as shown in table 4. We also present information on registered pregnancy complications in supplementary table S3. 

Information on relevant lifestyle factors such as BMI and substance abuse (e.g. smoking) are not available in the database. Information on maternal age and a limited number of other influencing factors are available and are presented in supplementary table S3, as mentioned in line 232.

• Table 3 Confidence interval are missing which impairs comparison

We have added confidence intervals in table 3 and 4. 

• Discussion: The ARRIVE trial should be discussed

We added a paragraph in the discussion on the difference between our results regarding caesarean sections and the results of RCTs on IOL vs expectant management (line 310-319) and a line on the possible effect on attitudes and beliefs regarding IOL (line 351-352). 

We are aware that results of observational research does not always reflect the results in RCTs. In case of the ARRIVE trial, we think that this study cannot be generalized to the Dutch setting. In this revised version of the article, we discuss this in line 315-319: “However, the ARRIVE trial reported a caesarean section rate of 18.6% in the IOL group vs. 22.2% in the expectant management group, which is considerably higher than the caesarean section rate observed in our study groups. Furthermore, the ARRIVE trial might suffer from selection bias. It is therefore questionable whether the results of the ARRIVE trial can be generalized to the Dutch setting.”

---

## [Editor Report · Decision Letter 1]

25 May 2023

Regional practice variation in induction of labor in the Netherlands: does it matter? A multilevel analysis of the association between induction rates and perinatal and maternal outcomes

PONE-D-22-29662R1

Dear Dr. Offerhaus,

We’re pleased to inform you that your manuscript has been judged scientifically suitable for publication and will be formally accepted for publication once it meets all outstanding technical requirements.

Kind regards,

David Desseauve, MD, MPH, PhD

Academic Editor

PLOS ONE

---

## [Editor Report · Acceptance letter]

30 May 2023

PONE-D-22-29662R1 

Regional practice variation in induction of labor in the Netherlands: does it matter? A multilevel analysis of the association between induction rates and perinatal and maternal outcomes 

Dear Dr. Offerhaus:

I'm pleased to inform you that your manuscript has been deemed suitable for publication in PLOS ONE. Congratulations! Your manuscript is now with our production department. 

Kind regards, 

on behalf of

Dr. David Desseauve 

Academic Editor

PLOS ONE